# The Structure and Properties of Laser-Cladded Inconel 625/TiC Composite Coatings

**DOI:** 10.3390/ma16031265

**Published:** 2023-02-01

**Authors:** Tomasz Poloczek, Aleksandra Lont, Jacek Górka

**Affiliations:** Welding Department, Faculty of Mechanical Engineering, Silesian University of Technology, Konarskiego Street 18A, 44-100 Gliwice, Poland

**Keywords:** composite coatings, nickel-based superalloys, laser cladding

## Abstract

This article presents production results concerning metal matrix composite-coatings made using the laser-cladding technology. The enhancement of the wear resistance of the material surface is the one of the main goals accompanying the manufacturing of composite coatings. Nickel-based superalloys are used in several industries because they are characterized by a number of desirable properties including high tensile and fatigue strength as well as resistance to high-temperature corrosion in aggressive environments. One of the most interesting materials from the group of superalloys is Inconel 625, used as a matrix material in tests discussed in this article. However, nickel-based superalloys are also characterized by an insufficient wear resistance of the surface, therefore, in relation to the tests discussed in this article, Inconel 625-based composite coatings were reinforced by adding 10%, 20% and 40% of titanium carbide particles. The addition of hard phases, i.e., TiC, WC or SiC particles can have a positive effect on the erosion resistance of cladded specimens. The aim of the experiment was to determine the impact of the titanium carbide content on the structure of the alloy and its resistance to corrosive wear, enabling the extension of the service life of Inconel 625/TiC composite coatings. The investigation included microhardness tests, corrosion resistance analysis, penetrant tests, macrostructure and microstructure analyses and X-ray diffraction (XRD) tests. The TiC particles increased the hardness of the coatings and, in general, had a negative impact on the corrosion resistance of pure Inconel 625 coatings. However, the increased homogeneity of composite coatings translated into the improvement of corrosion resistance.

## 1. Introduction

Surface engineering technologies are widely used to enhance surface properties such as corrosion or wear resistance without compromising favorable properties of the processed material core (e.g., mechanical and plastic properties) [1,2]. Nickel-based superalloys are widely used as coating materials due to their high corrosion resistance in aggressive environments as well as because of their fatigue strength, high-temperature toughness and ductility [3,4]. The positive influence of Inconel 625 coatings on the corrosion resistance of various grades of low-alloy and stainless steels was reported in the publication [5].

The improvement of erosion and tribological wear resistance of the surface entails the use of metal matrix composite (MMC) coatings or surface layers reinforced with ceramic particles [6,7]. The fabrication of composite coatings with the nickel-based superalloy matrix has a high application potential due to numerous advantageous properties combined with increased erosion and tribological wear resistance [8]. The results of previous research in this area revealed that the production of nickel-based superalloy coatings reinforced with carbides (WC [9], Cr_2_C_3_ [10], VC [11], TiC [12]), BN [13,14], TiB_2_ [15], carbon nanotubes [16] and graphene nanoplatelets [17] provides the enhancement of mechanical properties and wear resistance in comparison with those of metallic coatings. As regards the production of nickel-based MMC coatings, characterized by high-temperature corrosion resistance, it is reasonable to apply a reinforcing material of excellent thermal stability [18]. A ceramic material commonly used as composite reinforcement and characterized by thermal stability and a high melting point (of approximately 3180 °C) as well as high hardness (restricted within the range of 2859 HV to 3200 HV), strength (restricted within the range of 240 MPa to −390 MPa) and low density of 4.93 g/cm^3^ is titanium carbide (TiC) [19]. Previous research revealed that homogeneous nickel-based superalloy composite coatings reinforced with TiC particles (of various contents) could be produced using the laser-cladding technology [20]. Cao and Gu’s research results [21] revealed that the Inconel 625-based laser-cladded coatings reinforced with a 2.5% addition of nano-TiC were characterized by higher hardness as well as lower coefficients of friction and wear in comparison with those of pure Inconel 625. Jiang et al. [22] investigated Inconel 625-based composite coatings reinforced with a 5% addition of nano-TiC particles. The use of the above-named particles resulted in an increase in hardness and modulus in relation to metallic Inconel 625 coatings. Lian et al. [23] produced Ni35A-based laser-cladded coatings reinforced with an addition of between 20% and 80% of TiC particles. The research results showed that the increase in the TiC powder ratio resulted in an increase in hardness and wear resistance. Based on research results, it is obvious that Inconel 625-based composite coatings reinforced with TiC particles are characterized by higher wear and erosion resistance as well as hardness than those of coatings made of Inconel 625 without reinforcing phases. The addition of TiC phases to Inconel 625 coatings results in the microstructure changes, which could influence corrosion resistance. As regards the production of coatings, operated under conditions of surface wear and exposed to an aggressive corrosive environment, it is very important to produce wear-resistant coatings in which the addition of reinforcements will not worsen corrosion resistance. Ge et al. [24] conducted research on the corrosion resistance of Inconel 625-based coatings reinforced with a 4% addition of TiC particles in relation to laser-cladding process parameters. Bakkar et al. [25] tested the corrosion resistance of Inconel 625/TiC composites produced using the squeeze-casting technique. The results revealed that an addition of 25% of TiC did not affect the corrosion resistance of Inconel 625, while higher contents resulted in the deterioration of corrosion resistance. A summary of prior works is presented in Table 1.

The main purpose of the study was to test the influence of the TiC particles’ addition to Inconel 625 laser-cladded coatings on their corrosion resistance, as the metallic Inconel 625 coatings are commonly used in aggressive corrosive environments. Generally, the composite coatings are produced for wear-resistance improvement. The erosive wear-resistance of the laser-cladded Inconel 625 coatings reinforced by 10–40% of TiC particles was tested before by the authors, proving that the produced composite coatings showed better resistance to solid particle erosion in comparison to metallic Inconel 625 coatings. The current study was carried out to verify if the addition of TiC particles in those fractions influences the corrosion resistance of coatings. The research-related investigation involved the performance of penetrant tests, macrostructure and microstructure observations, chemical and phase composition analyses, microhardness tests and potentiodynamic polarization corrosion tests.

## 2. Materials and Methods

### 2.1. Materials and Laser Processing

The laser cladding of metallic Inconel 625 and composite Inconel625/TiC coatings was performed on substrate steel S355JR (Cognor, Stalowa Wola, Poland) having dimensions of 100 mm × 100 mm × 10 mm. Before the laser-cladding process, specimens were subjected to grinding and degreased using ethyl alcohol (Stanlab, Lublin, Poland). For the purpose of the laser-cladding process, Inconel 625 (Metcoclad 625 gas atomized powder, Oerlikon, Westbury, NY, USA) and 99.8% pure TiC (TI546030/2; Goodfellow, Huntington, UK) powders were mixed in Inconel 625 to TiC volume ratios of 100:0, 90:10, 80:20 and 60:40. The TiC particles size was 50–150 µm. Before the laser-cladding process, in order to remove moisture from the powder, the material was dried for 1 h at a temperature of 50 °C. The chemical compositions of the substrate material and of the powder Inconel 625 are listed in Table 2.

The testing stand used in laser processing was equipped with a solid-state laser (TRUMPF, Ditzingen, Germany), a numerical positioning-control system and a powder feeder. The laser beam focus (having a diameter of 200 µm) was set 30 mm above the substrate surface. The powder was injected directly into the melt pool. Argon was used both as powder transporting and shielding gas (flow rates being 3 L/min and 10 L/min, respectively). The coatings were made using the multi-run of single passes with a 40% overlap. The laser processing was performed without preheating; the interpass temperature amounted to less than 30 °C. Process parameters are shown in (Table 3) and were adjusted taking into consideration the previous experience demonstrated in [26].

### 2.2. Penetrant Tests and Macroscopic Investigation

Subsequent penetrant tests were conducted in the color contrast technique using penetrant 68 NF, developer MR 70 and cleaner MR 79 (MR Chemie, Unna, Germany). Both dwell and development times applied in the penetrant tests were 10 min in duration. The macrostructure analysis included macrostructure observations performed using a Phenom World PRO scanning electron microscope (SEM) (Thermo Fisher Scientific, Walham, MA, USA) featuring quality and homogeneity assessment. The coating dilution was calculated using Equation (1), where *F_BM_* is the parameter of the cross-sectional area of the melted substrate and *RA* is the parameter of cross-sectional area of the coating. Measurements were performed using an AutoCAD 2020 software program (Autodesk, CA, USA). The TiC particle volume fraction was calculated using an Image-Pro Plus software program (Media Cybernetics, Inc., Rockville, MD, USA).
(1)$$U=\frac{F_{BM}}{F_{BM}+RA}\times 100\;\left[\%\right]$$

### 2.3. Microstructure Investigation and Hardness Testing

The energy dispersive spectroscopy-based (EDS) chemical composition analysis and microstructure observations were performed using the Phenom World Pro scanning electron microscope (SEM). Specimens used in macrostructure and microstructure observations were cut out 50 mm away from the laser-processing initiation area (i.e., the area where the process was characterized by appropriate stability). The etching process involved the use of the mixture of HNO_3_ (Chempur, Piekary Śląskie, Poland), HCl (Chempur, Piekary Śląskie, Poland), acetic acid (Stanlab, Lublin, Poland) and glycerol (Poch, Gliwice, Poland) (etchant 89 according to ASTM E 407-99); the specimens were heated by being immersed for 2 min in distilled water having a temperature of 100 °C. Afterwards, the specimens were etched through immersion in the reagent for 10 s. Based on the four cross-sectional surface, the EDS data obtained at a magnification of 1000× and an acceleration voltage of 15 kV it was possible to determine the general chemical composition of the coatings. The X-ray diffraction (XRD) test was performed using a PANanalytical X’Pert PRO (Malvern Panalitycal, Malvern, UK) diffraction system equipped with a cobalt anode. The diffraction profiles were obtained in a continuous scan mode within the 2ϴ range of 25° to 130°, a step size of 0.1444° and a counting time per step of 0.026 s.

The Vickers microhardness of the laser-cladded coatings was measured using a Wilson 401MVD (Wilson Instruments, Instron Company, Norwood, MA, USA) tester, a load of 200 g and a dwell time of 12 s. The tests were carried out in three lines across the coating, i.e., 0.7 mm, 1.0 mm and 1.3 mm away from the surface (Figure 1a); the distance between measurement points being 0.5 mm. In addition, the microhardness tests were performed in three lines from the surface to the base material, with a distance of 0.1 mm between neighboring points (Figure 1b).

The corrosion behavior of the metallic and composite laser-cladded coatings was tested using potentiodynamic polarization experiments in accordance with the ISO 17475:2010 standard. The tests involved the use of an Autolab 302 N potentiostat (nLab, Warszawa, Poland) equipped with a three-electrode cell controlled by a NOVA software program. Electrochemical measurements were conducted in a 3.5% NaCl solution at a temperature of 25 °C. A saturated calomel electrode (SCE) was used as a reference electrode, whereas a platinum rod was used as a counter electrode. The corrosion resistance parameter was assessed by recording a change in the open-circuit potential (EOCP) in relation to the saturated calomel electrode. Before the tests, coating surfaces were prepared by grinding. The specimens were evaluated after 600 s of open-circuit potential stabilization at a scan rate of 1 mVs^−1^. The surface morphology after the electrochemical tests was analyzed using a Zeiss EVO MA10 scanning electron microscope (SEM) (ZEISS, Jena, Germany).

## 3. Results and Discussion

### 3.1. Penetrant Tests

The penetrant tests revealed linear indications triggered by the presence of cracks on two coatings obtained using the powder mixture having a TiC particle fraction of 40 vol.% (Figure 2). No indications were observed on the surfaces of the metallic and those of the remaining composite coatings. The cracks on the coatings with the highest TiC particle fraction were induced by the higher hardness and brittleness of these phases in combination with different thermal expansion coefficient between TiC and Inconel 625, resulting in weak interface bonding between these phases.

### 3.2. Macrostructure

The macrographs of the metallic and composite coatings are presented in Figure 3, the parameters of thickness, dilutions and measured TiC fractions are summarized in Table 4, whereas the average chemical compositions of the coatings obtained in the EDS analysis are presented in Table 5. The test results made it possible to determine the influence of the powder feed rate and that of the TiC ratio on the thickness, dilution, penetration and the uniformity of reinforcing phase-particle distribution in the microstructure of the coatings. The macrograph observations enabled the assessment of the quality of the coatings. In the case of all the laser-cladded coatings made using a powder feed rate of 0.04 g/mm, no imperfections were observed in the cross-sections. The macroscopic observations also revealed that the uniformity of TiC distribution improved along with an increased TiC fraction. The coatings with TiC contents of 10 vol.% and 20 vol.% contained clusters of reinforcing particles. The presence of the above-named clusters, observed mainly in the upper area of the coatings, was caused by the lower density of the TiC particles than those of Inconel 625. The increased powder feed-rate resulted in the lower penetration and dilution of the substrate, which resulted in the local lack of fusion of the coatings obtained using the powder mixtures having a TiC content of 20 vol.% and that of 40 vol.%. In the composite coatings obtained using constant parameters, dilution decreased along with an increasing TiC content. However, the lowest dilution was obtained in the metallic Inconel 625 coatings. Higher dilution observed in the composite coatings could be ascribed to the higher value of laser radiation absorption, which, in turn, could be attributed to the presence of the titanium carbide phase in the powder. On the other hand, the TiC ratio increase led to decreased penetration, which could result from the Marangoni convection inhibition [27]. The thickness of the coatings slightly increased along with an increase in the powder feed rate and the TiC ratio. In relation to each composite coating, the measured TiC fraction was lower than the TiC content in the prepared powder mixtures (which was related to dilution). In each coating, the TiC fraction was higher when the dilution was lower. The higher dilution led to an increase in the average iron content (Table 5).

### 3.3. Microstructure

The microstructure of the TiC-reinforced composite Inconel 625 coatings is presented in Figure 4. Figure 5 shows the microstructure of the metallic Inconel 625 coating. The matrix microstructure of the composite coatings consists of austenite dendrites (the presence of which was confirmed by XRD results, see Figure 6) and minor secondary carbide phases. According to the EDS analysis (Figure 7), the austenite dendrites are mainly composed of nickel, chromium and iron, whereas the secondary phases are characterized by the high concentration of carbon, niobium, molybdenum and titanium. The chemical composition of the Metcoclad 625 powder does not contain titanium or carbon. Therefore, the secondary phases could be attributed to the partial dissolution of TiC in the matrix and its enrichment with carbon and titanium. The foregoing was confirmed by the observations of the metallic Inconel 625 coating microstructure, where no such precipitates were found. The coatings without reinforcing phases revealed the typical dendritic structure with minor constituents in the interdendritic regions (already reported by Cieslak et al. [28]). In the case of the metallic Inconel 625 coatings, the columnar growth of dendrites was triggered by temperature gradient during solidification, with the dendrites growing in the direction opposite to the heat transfer.

The secondary phases formed in the matrix, characterized by blocky and dendritic morphology, revealed the gradient distribution of chemical contents (Figure 4). The EDS examination revealed that the inner, i.e., darker, part of the aforesaid phases contained a higher titanium content. It could be assumed that the TiC particles precipitated from the liquid metal and, afterwards, the atoms of Mo and Nb dissolved in the crystal lattice. Eutectic phases, which precipitated on the secondary phases, could be observed in the matrix. After the formation of the secondary phases, they served, during cooling, as the crystal nucleus for eutectic precipitates between the austenite dendrites.

In the microstructure of the overlap areas located between consecutive beads (Figure 8), the austenite dendrites as well as the secondary and the eutectic phase precipitates could be observed in the matrix (as in the central bead areas). In addition, it was also possible to observe large dendritic precipitates, formed as a result of the higher TiC particle dissolution triggered by the subsequent thermal cycle. In the aforesaid area, the TiC particles were more rounded and characterized by the lighter shell (composed of carbon, niobium, molybdenum and titanium—the same as in the dendritic precipitates (Figure 9)). The morphology of the dendritic precipitates was characteristic of in situ TiC particles [29]. Therefore, it can be assumed that the TiC precipitates were formed in the overlap area, whereas the Nb and Mo atoms dissolved in their crystal lattice.

### 3.4. Hardness

The Vickers microhardness test results are presented in Figure 10. The average microhardness of Inconel 625 reinforced with TiC was restricted within the range of 258 HV0.2 to 342 HV0.2. The addition of between 10 vol.% and 40 vol.% of titanium carbide particles triggered an increase in average microhardness of between 5% and 50% in comparison with that of the Inconel 625 metallic coatings. An increase in the TiC particle content was accompanied by an increase in the microhardness of the coatings subjected to the laser-cladding process performed using constant parameters. In turn, an increase in the powder feed rate during the laser-cladding process carried out using powder having the same TiC particle fraction triggered an increase in average microhardness, which was related to the lower dilution and higher fraction of TiC particles. In each of the test coatings, no significant variations of microhardness parameters were observed in the overlap area located between consecutive beads. A minor microhardness decrease was observed towards the end of the measurement line (Figure 10b), which was connected with the higher dilution of the first pass (see Figure 3). The microhardness distribution from the coating surface to the base material (Figure 10c) revealed the highest hardness on the surface area and a slight decrease towards the base material. The above-named phenomenon was related to a higher fraction of carbides on the surface area of the coating, where TiC, characterized by lower density than that of the matrix material, caused the former to flow upwards in the molten metal pool.

### 3.5. Corrosion Behavior

The potentiodynamic polarization curves obtained for metallic Inconel 625 coating M1 and TiC-reinforced Inconel 625 coatings C1, C3 and C5 are presented in Figure 11. The corrosion results are summarized in Table 6; the values of corrosion potential (E_corr_) and corrosion current density (j_corr_) were obtained using the Tafel extrapolation method. In general, the materials characterized by a high corrosion potential and low corrosion current density were also characterized by a high corrosion resistance [12]. The values of E_corr_ and j_corr_ in relation to the metallic Inconel 625 coating amounted to −0.384 V and 8.0 µA/cm^2^, respectively. In terms of the composite coatings, the corrosion potential was restricted within the range from −0.377 V to −0.473 V, whereas the corrosion current density was between 8.9 µA/cm^2^ and 83.0 µA/cm^2^. The results concerning corrosion current density revealed that the highest corrosion resistance was obtained in relation to the metallic Inconel 625 coating, whereas the composite coating containing a TiC fraction of 38.6 vol.% (C5) was characterized by slightly lower corrosion resistance. The lowest corrosion resistance was that of composite coating C3 (TiC fraction of 18.3 vol.%), characterized by 10-fold higher corrosion current density than that of the metallic coating. The results of corrosion potentials, related to the composite coatings revealed that the most favorable results were those of the coating having the highest TiC fraction (C5); the lowest corrosion resistance was observed in coating C3. The metallic Inconel 625 coating was characterized by a slightly lower corrosion potential than that of composite coating C5. In general, both materials (Inconel 625 alloy [30] and TiCs [31]) offer high corrosion resistance in the above-presented solution, yet the microstructure of the composite coatings may affect their corrosion resistance as well as other characteristics such as uniformity, grain size, particle-matrix interface quality and the presence of intermetallic phases [24]. Among the test composite coatings, coating C5 was characterized by the highest corrosion resistance (similar to that of the metallic Inconel 625 coating), resulting from the highest structure homogeneity [32].

The morphologies of corrosion damage affecting the metallic and composite coatings are presented in Figure 12. Observations of the surface morphology after the corrosion tests revealed the presence of corrosion pits on both the metallic and composite coatings. In addition, in the case of coatings C1 and C3, it was possible to observe the passive film peeling, which indicated insufficient corrosion resistance in the NaCl solution (stability of the passive film determines the corrosion resistance of materials). Among the test composite coatings, the lowest damage was observed in coating C5; no passive film peeling was observed. However, Inconel 625 was characterized by the lowest surface corrosion damage among all the coatings subjected to the tests.

The above-presented test results revealed that the metallic Inconel 625 coatings were characterized by higher corrosion resistance than that of the composite coatings reinforced with TiC particles. However, the ensuring of the high homogeneity of the composite coating structure enabled the obtainment of corrosion resistance similar to that of Inconel 625.

## 4. Conclusions

The research concerning the laser-cladding process involving the use of Inconel 625 powder reinforced with TiC particles allowed the formulation of the following conclusions:

The process of laser cladding can be used for the fabrication of homogeneous Inconel 625-based ex situ composite coatings containing up to 40 vol.% of TiC particles. The parameters of the powder feed rate, constant laser beam power and cladding rate influenced the penetration and dilution of coatings; however, it had no significant influence on their homogeneity.During the laser-cladding process, the TiC particles partially dissolved in the liquid pool, thus causing enrichment of the matrix with carbon and titanium. The crystallization process was accompanied by the precipitation of minor blocky secondary phases rich in carbon, titanium, niobium and molybdenum. The dissolution of TiC particles was more intense in the overlap area and triggered the additional formation of dendritic precipitates rich in carbon, titanium, niobium and molybdenum.The average microhardness of the composite Inconel 625/TiC coatings was higher than that of the metallic Inconel 625 coatings and was restricted within the range of 258 HV0.2 to 342 HV0.2. The average microhardness of the coatings increased along with an increase in the TiC content. The increase in coating dilution, accompanied by a decrease in the powder feed rate resulted in a decrease in average microhardness. In the overlap areas, no significant or repeated microhardness changes were noticed.The Inconel 625 coating was characterized by high corrosion resistance in a 3.5% NaCl solution. The corrosion resistance of the composite TiC-reinforced coatings was lower than that of the metallic coating.

## Figures and Tables

**Figure 1 materials-16-01265-f001:**
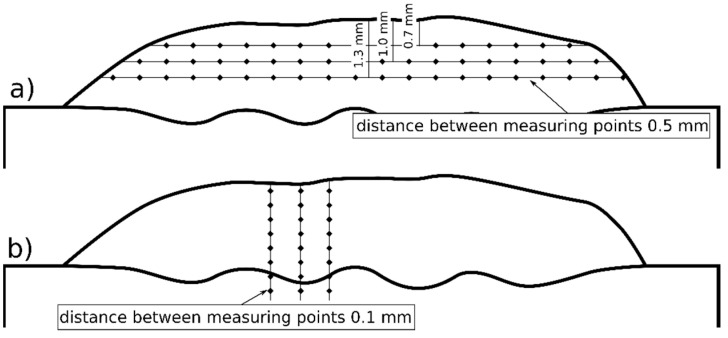
Schematic diagram of Vickers microhardness measurement lines: (**a**) measurements across the beads and (**b**) measurements from the surface to the base material.

**Figure 2 materials-16-01265-f002:**
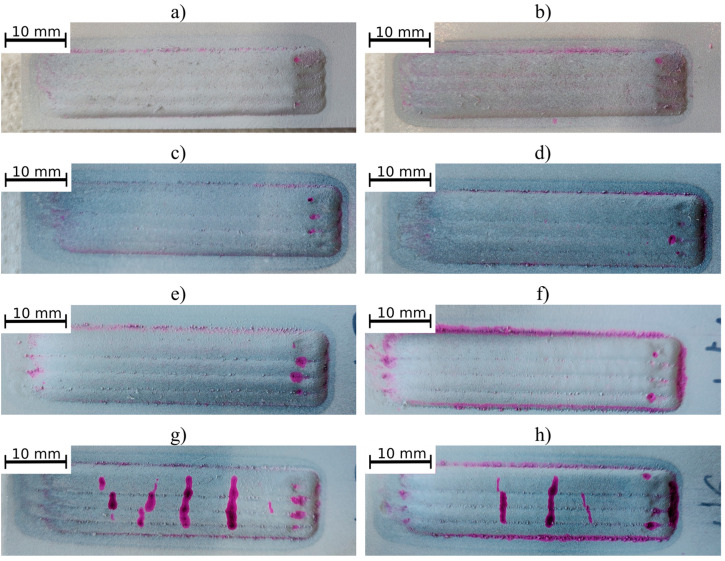
Surface of the coatings after penetrant testing: (**a**) M1, (**b**) M2, (**c**) C1, (**d**) C2, (**e**) C3, (**f**) C4, (**g**) C5 and (**h**) C6, designations in accordance with Table 3.

**Figure 3 materials-16-01265-f003:**
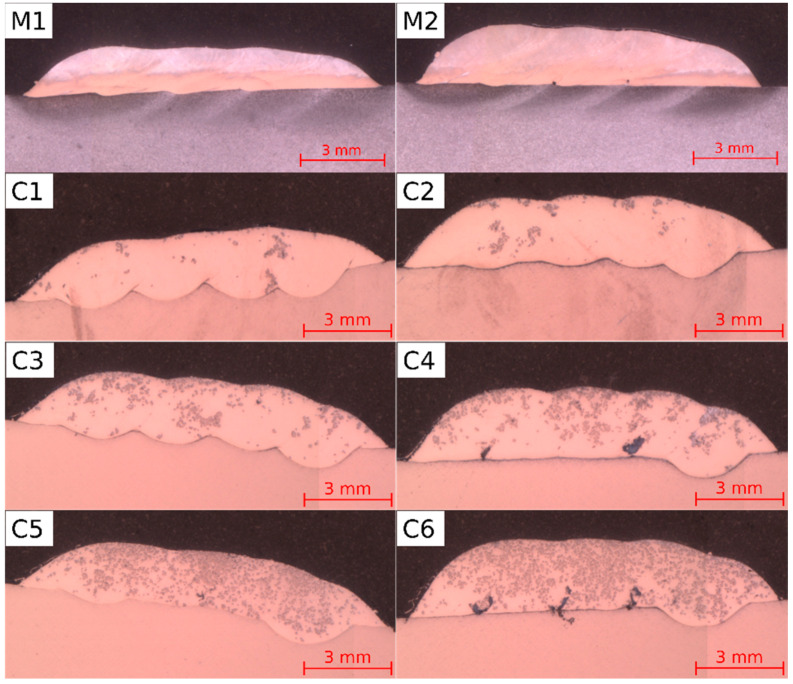
Macrographs of the laser-cladded coatings; designations in accordance with Table 3. Samples M1, M2 made without TiC, samples C1–C6 made with the addition of TiC.

**Figure 4 materials-16-01265-f004:**
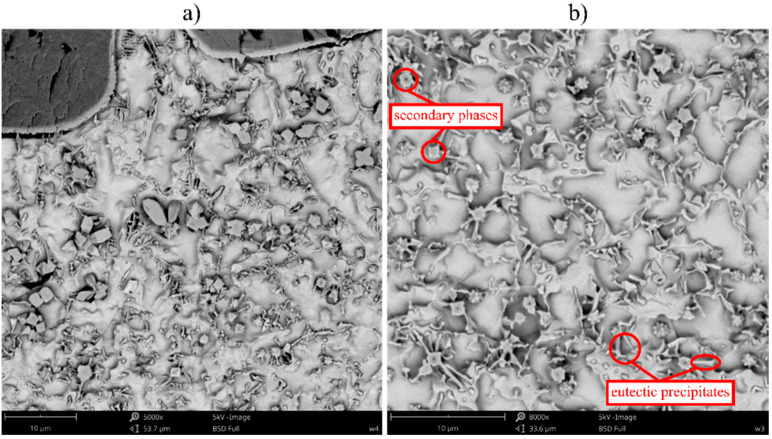
Microstructure of the central bead areas of the composite Inconel 625/TiC coatings: (**a**) C4, (**b**) C3, designations in accordance with Table 3.

**Figure 5 materials-16-01265-f005:**
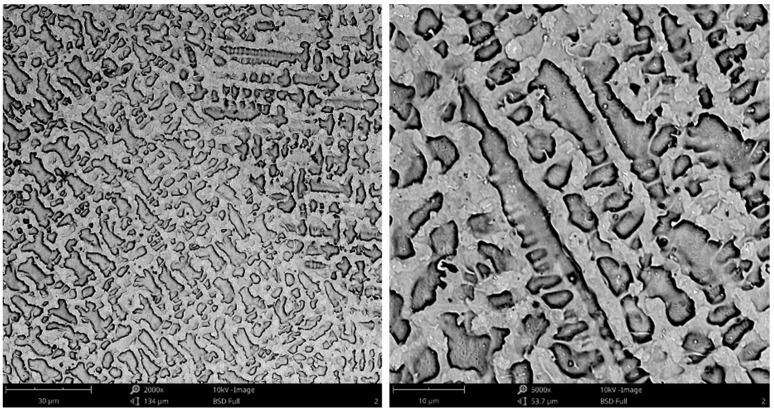
Microstructure of the representative metallic Inconel 625 coating.

**Figure 6 materials-16-01265-f006:**
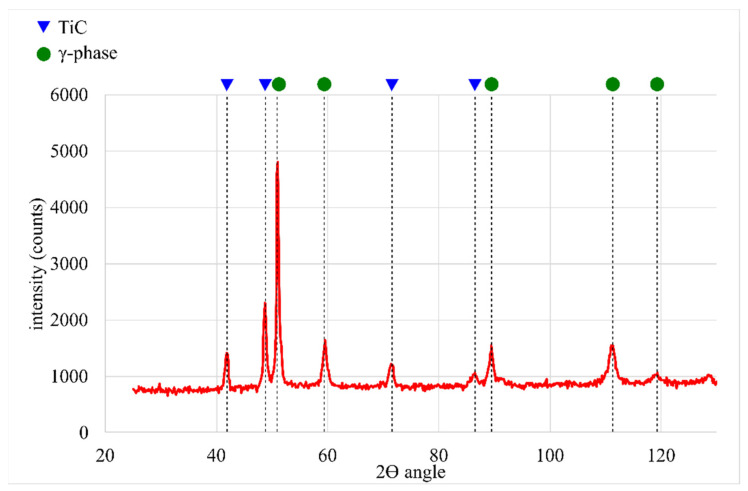
XRD results of the representative composite Inconel 625/TiC coating.

**Figure 7 materials-16-01265-f007:**
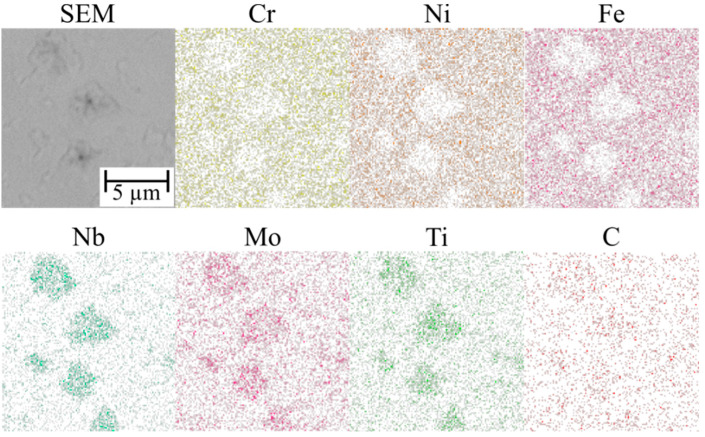
EDS maps of the composite Inconel 625/TiC coating matrix.

**Figure 8 materials-16-01265-f008:**
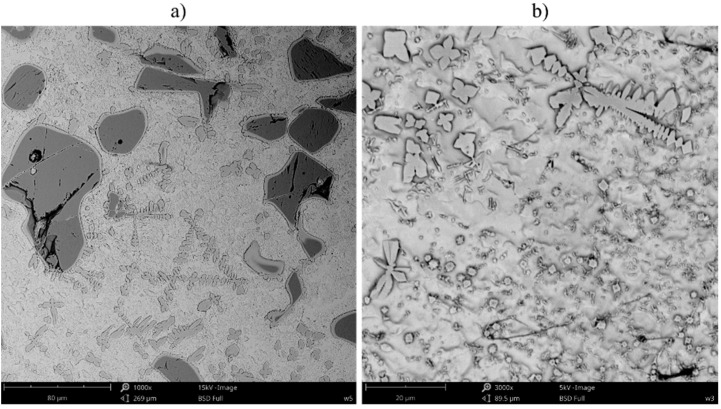
Microstructure of the overlap areas of the composite Inconel 626/TiC coatings; magnification (**a**) 1000×, (**b**) 3000×.

**Figure 9 materials-16-01265-f009:**
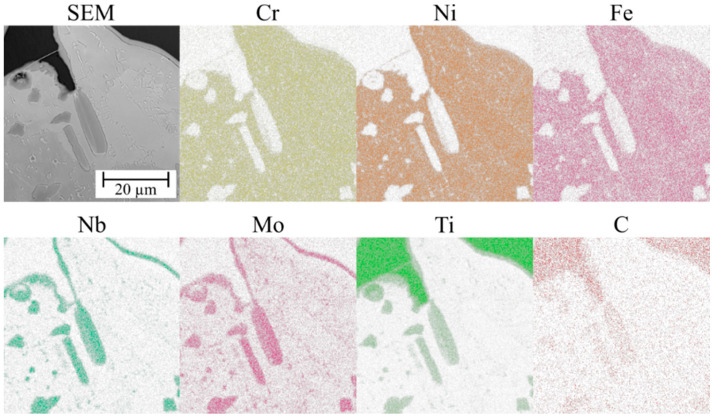
EDS maps of the overlap area of the composite Inconel 625/TiC coatings.

**Figure 10 materials-16-01265-f010:**
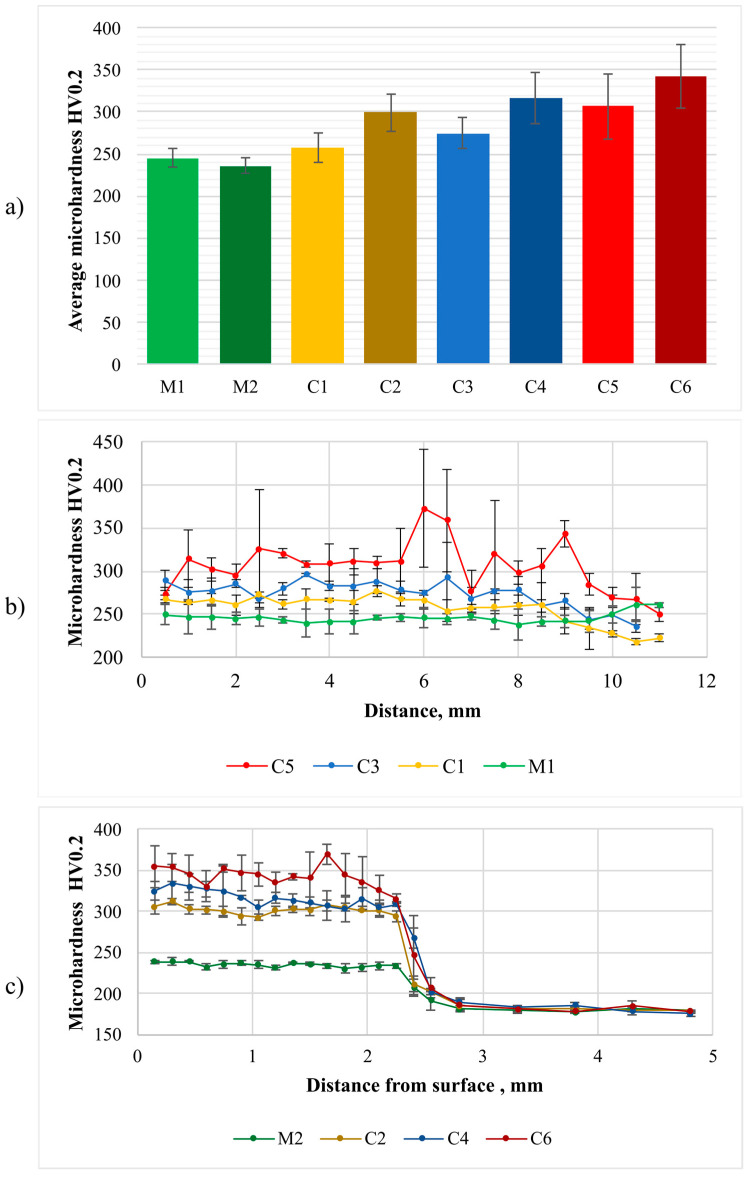
Vickers microhardness test results: (**a**) average microhardness, (**b**) microhardness distribution across the beads (in accordance with Figure 1a) and (**c**) microhardness distribution from the surface to the base material (in accordance with Figure 1b); designations in accordance with Table 3.

**Figure 11 materials-16-01265-f011:**
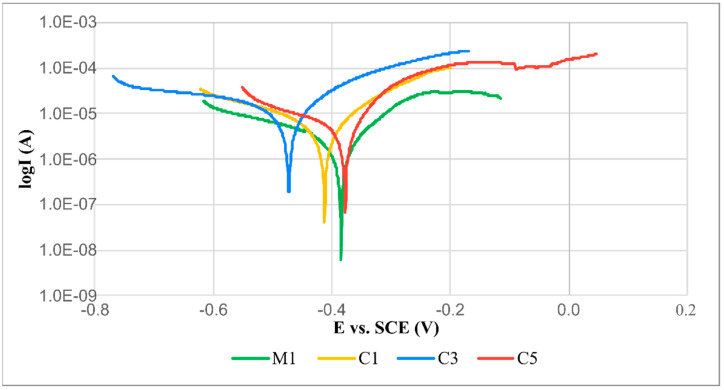
Potentiodynamic polarization curves of laser-cladded coatings M1, C1, C3 and C5; designations in accordance with Table 3.

**Figure 12 materials-16-01265-f012:**
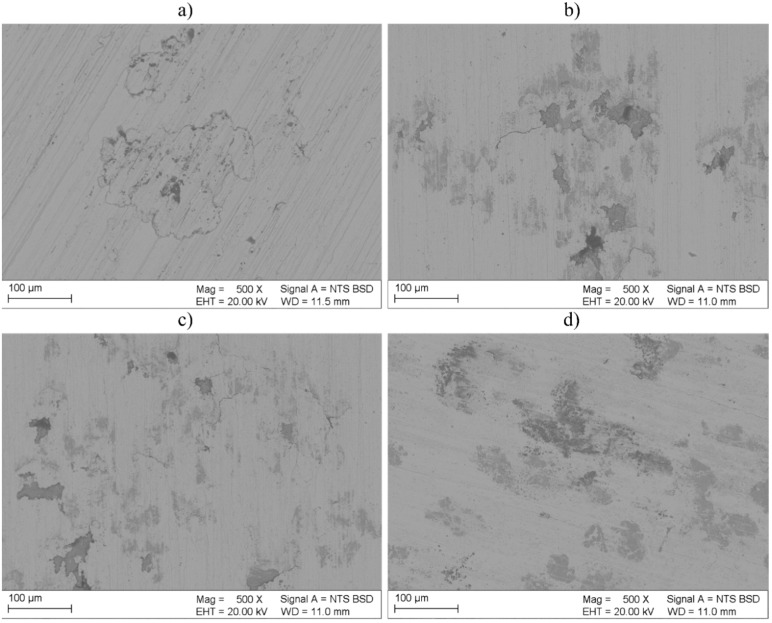
Morphologies of corrosion damage after potentiodynamic polarization tests, SEM, coatings: (**a**) M1, (**b**) C1, (**c**) C3 and (**d**) C5 designations in accordance with Table 3.

**Table 1 materials-16-01265-t001:** Chemical compositions of S355JR substrate material and Metcoclad 625 powder.

Authors	Material	Results in Comparison to Material without Reinforcing Phase
Gopinath et al. [20]	Inconel 718 + 30–70% TiC	higher brittleness
Cao and Gu [21]	Inconel 625 + 2.5% nano-TiC	higher hardness, lower coefficient of friction lower wear rate
Jiang et al. [22]	Inconel 625 + 5% nano-TiC	higher hardness, higher Young modulus
Lian et al. [23]	Ni35A + 20–80% TiC particles	increase in TiC ratio resulted in higher hardness and wear resistance
Ge et al. [24]	Inconel 625 + 4% TiC	carbides played a role in precipitation strengthening and enhancing the anti-corrosion properties
Bakkar et al. [25]	Inconel 625 + 25%/50%/70% TiC	higher hardness, corrosion behavior comparable to monolithic Inconel 625

**Table 2 materials-16-01265-t002:** Chemical compositions of substrate material S355JR and powder Metcoclad 625.

Material	C	Mn	Si	P	S	Fe
S355JR (wt.%)	0.2	1.5	0.2–0.5	max 0.04	max 0.04	balance
	Cr	Ni	Mo	Nb	Fe	
Oerlikon Metcoclad 625 (wt.%)	20.0–23.0	58.0–63.0	8.0–10.0	3.0–5.0	max 5.0	

**Table 3 materials-16-01265-t003:** Laser-cladding process parameters.

Designation	TiC Powder Content (vol.%)	Laser Power (W)	Cladding Rate (m/min)	Powder Feed Rate (g/mm)
M1	0	2100	0.25	0.04
M2	0			0.05
C1	10			0.04
C2	10			0.05
C3	20			0.04
C4	20			0.05
C5	40			0.04
C6	40			0.05

**Table 4 materials-16-01265-t004:** Thickness, dilution and TiC volume fraction measurement results; designations in accordance with Table 3.

Designation	Thickness (mm)	Dilution (%)	Measured TiC Volume Fraction (vol.%)
M1	1.6	3.3	-
M2	2.1	2.1	-
C1	1.7	25.5	8.8
C2	2.1	12.6	9.8
C3	1.8	17.5	18.3
C4	2.2	9.8	19.6
C5	1.9	14.6	38.6
C6	2.3	7.5	39.7

**Table 5 materials-16-01265-t005:** Average chemical composition of the coatings, designations in accordance with Table 3.

Designation	Ni	Cr	Mo	Nb	Fe	Ti
M1	60.7 ± 1.6	19.8 ± 0.5	10.2 ± 0.8	4.6 ± 0.1	4.7 ± 1.1	-
M2	63.6 ± 0.6	20.7 ± 0.3	9.5 ± 0.5	4.4 ± 0.6	1.8 ± 0.3	-
C1	49.1 ± 1.1	16.2 ± 0.4	8.1 ± 0.6	4.7 ± 0.4	18.0 ± 1.0	2.7 ± 0.5
C2	54.1 ± 4.7	17.6 ± 1.4	9.3 ± 1.1	4.2 ± 0.9	7.9 ± 2.6	3.5 ± 1.1
C3	50.3 ± 1.3	16.6 ± 0.2	9.7 ± 1.4	3.9 ± 0.3	14.4 ± 0.8	5.4 ± 2.1
C4	55.1 ± 1.9	18.1 ± 0.5	10.1 ± 1.3	5.1 ± 0.2	5.1 ± 1.6	6.4 ± 1.3
C5	44.1 ± 2.9	14.1 ± 2.2	7.4 ± 0.9	4.3 ± 0.5	14.5 ± 3.9	14.6 ± 3.8
C6	47.0 ± 4.9	16.0 ± 1.8	8.6 ± 1.4	4.8 ± 0.8	3.3 ± 1.9	19.7 ± 8.5

**Table 6 materials-16-01265-t006:** Electrochemical parameters of the laser-cladded coatings, designations in accordance with Table 3.

Designation	c_ord_ (µA/cm^2^)	E_corr_ (V)
M1	8.0	−0.384
C1	12.0	−0.412
C3	83.0	−0.473
C5	8.9	−0.377

## Data Availability

Data sharing not applicable.

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
