# Peer review of "The Structure and Properties of Laser-Cladded Inconel 625/TiC Composite Coatings"

_materials, 2023, doi:10.3390/ma16031265_

Round 1

Reviewer 1 Report

The authors studied the microstructure, properties and corrosion behavior of laser cladding Inconel 625/TiC composite coatings. The results in this work are interesting. Comments and suggestions for authors:

1.     Please supplement the penetration test pictures of M1, M2 and C1-C4 samples.

2.     Line 155: The cause of the crack cannot be attributed simply to brittleness. Please provide more evidence.

3.     The content of TiC will affect the coating performance, so what is the effect of TiC content on coating microstructure formation?

4.     The properties of the coatings in this work should be compared with the other  similar MMC coatings.

5.     It is not indicated in Fig.3 that the uniformity of the coating can be improved with increase of the proportion of TiC. The TiC particles in C5/C6 samples are still concentrated in the upper part of the coating.

6.     The addition of TiC cannot significantly improve the corrosion resistance of composite coating, so what is the significance of composite coating.

7.     Please explain why the corrosion resistance of composite coatings is inferior to that of Inconel 625 coatings.

8.     Please check the citation format in reference [27].

9.     Line 372: The page number of the document cited is incorrect.

Author Response

Dear Reviewer,

thank you very much for your time and valuable comments contained in the article, which we hope will increase its scientific value. Please find attached the responses to the review.

Seriously,

Authors

Reviewer 2 Report

Comments to Authors:

Ref. No.: materials-2143756

Title: The structure and properties of laser cladded Inconel 625/TiC composite coatings

Overview and general recommendation:

In this study, the authors used laser cladding technology to manufacture composite coatings of Inconel 625 reinforced with 10%, 20%, and 40% of TiC particles. Then the authors studied how the parameter of powder feed rate can affect the homogeneity of Inconel 625-based ex-situ composite coatings, based on their microstructure. They also studied and compared the corrosion resistance of Inconel 625 coating and composite TiC-reinforced coatings. However, there still remain some problems that the authors need to address first before publication.

Major comments:

1.     The main aim of this study is not very clear to readers. Please explain clearly why the authors performed this study in the Introduction section.

2.     The Abstract section is not well written. The authors failed to briefly explain what the aim is for this study, why it is important, how the authors address the challenges, and what the results are like.

3.     In 2. Materials and Methods and 3. Results and discussion, if the authors can divide these sections into some subsections, it will help the authors better to organize their content. In the current version, everything is messed up together. It makes it very hard for the readers to follow the authors’ discussion.

4.     When the authors discussed the corrosion resistance, the authors mentioned that the materials that show high corrosion potential and low corrosion current density are characterized by high corrosion resistance. Then in Table 2, how could the authors compare the corrosion resistance just based on one single parameter? If two parameters have to be considered together, what are the criteria for the comparison?

5.     There are still many grammar errors in the manuscript. This makes it very hard to read and understand the materials presented in the manuscript. The authors really read to address this first before submitting their work for review.

Author Response

(The authors gave the same response as above.)

Reviewer 3 Report

The following comments need to be addressed

1) Table 1 is unclear to read

2) Please make a table of prior works by researchers followed by the novelty of the present work. this gives lot of clarity and information to the reader

3) the nomenclature of the specimen needs to be provided. its very difficult to understand the results without proper identification of the specimen. 

4) From Table 3, it is observed that the coated specimens are of different coating thicknesses which would result in different microhardness values. Why can't the thickness be made constant and investigate its effect on hardness

5) SEM images of the cross-section give better clarity on the structure of the coatings rather than micrographs using metallurgical microscope

Author Response

(The authors gave the same response as above.)

Round 2

Reviewer 2 Report

Thanks to the authors for their efforts to address my comments. Compared with the previous version, the grammar has improved a lot in the current version. But the authors still need to answer my question: what is the main aim and novelty of this study? Based on the authors' literature review, there are already some studies on Inconel 625-based coatings reinforced with TiC particles using a laser cladding process. Then why do the authors still need to conduct this study? Why is the loading of TiC at 10-40%? Do you have any particular reason for this? Another key question is about the property of TiC. What is their size? Are they nanoparticles?  Since this is a commercial product, the authors may need to perform some characterization tests on the materials first. 

Last but not least, could the author provide a clean version of the manuscript for review? With those marks, it takes a lot of work to read through the content. 

Author Response

Dear Reviewer,

Once again, thank you very much for taking the time to review our article, which undoubtedly increased its scientific value. Below are the answers to the comments in the second review.

Answers to comments:

  1. But the authors still need to answer my question: what is the main aim and novelty of this study? Based on the authors' literature review, there are already some studies on Inconel 625-based coatings reinforced with TiC particles using a laser cladding process. Then why do the authors still need to conduct this study? Why is the loading of TiC at 10-40%? Do you have any particular reason for this?

Answer: The main purpose of the study was to test the influence of the TiC particles addition to Inconel 625 laser cladded coatings on their corrosion resistance, as the metallic Inconel 625 coatings are commonly used in the aggressive corrosive environments. Generally, the composite coatings are produced for the wear resistance improvement. The erosive wear resistance of the laser cladded Inconel 625 coatings reinforced by 10-40% of TiC particles was tested before by the authors, proving that the produced composite coatings show better resistance to solid particle erosion in comparison to metallic Inconel 625 coatings. The current study was carried out to verify if the addition of TiC particles in those fractions influences the corrosion resistance of coatings. (Corrected in the text).

  1. Another key question is about the property of TiC. What is their size? Are they nanoparticles? Since this is a commercial product, the authors may need to perform some characterization tests on the materials first.

Answer: Thank you for the suggestion to test the powders before study. The TiC particles size is 50-150 µm. (Completed in the text).

  1. Last but not least, could the author provide a clean version of the manuscript for review? With those marks, it takes a lot of work to read through the content.

Answer: Of course, we realize that such a text is difficult to review (for which we apologize), but the requirements of the Publishing House enforce the change tracking mode. In the attachment we send the finished text after removing the introduced changes.

Regards,
Authors
